# Brief Oxygen Exposure after Traumatic Brain Injury Hastens Recovery and Promotes Adaptive Chronic Endoplasmic Reticulum Stress Responses

**DOI:** 10.3390/ijms24129831

**Published:** 2023-06-06

**Authors:** Jordyn N. Torrens, Shelby M. Hetzer, Nathan K. Evanson

**Affiliations:** 1Division of Pediatric Rehabilitation Medicine, Cincinnati Children’s Hospital Medical Center, Cincinnati, OH 45229, USA; jordyn.torrens@cchmc.org; 2Neuroscience Graduate Program, University of Cincinnati, Cincinnati, OH 45267, USA

**Keywords:** traumatic brain injury, traumatic optic neuropathy, endoplasmic reticulum stress, adolescent, neurodegeneration

## Abstract

Traumatic brain injury (TBI) is a major public health concern, particularly in adolescents who have a higher mortality and incidence of visual pathway injury compared to adult patients. Likewise, we have found disparities between adult and adolescent TBI outcomes in rodents. Most interestingly, adolescents suffer a prolonged apneic period immediately post-injury, leading to higher mortality; therefore, we implemented a brief oxygen exposure paradigm to circumvent this increased mortality. Adolescent male mice experienced a closed-head weight-drop TBI and were then exposed to 100% O_2_ until normal breathing returned or recovered in room air. We followed mice for 7 and 30 days and assessed their optokinetic response; retinal ganglion cell loss; axonal degeneration; glial reactivity; and retinal ER stress protein levels. O_2_ reduced adolescent mortality by 40%, improved post-injury visual acuity, and reduced axonal degeneration and gliosis in optical projection regions. ER stress protein expression was altered in injured mice, and mice given O_2_ utilized different ER stress pathways in a time-dependent manner. Finally, O_2_ exposure may be mediating these ER stress responses through regulation of the redox-sensitive ER folding protein ERO1α, which has been linked to a reduction in the toxic effects of free radicals in other animal models of ER stress.

## 1. Introduction

The association between brain oxygen deprivation (i.e., hypoxia) and traumatic brain injury (TBI) has been known for decades [1], with a reported coincidence of up to 44% [2]. Several studies have shown that brain tissue oxygen tension can serve as an independent predictor of TBI outcomes [3,4]. Indeed, guidelines for critical care after TBI are largely focused on maintaining brain perfusion and oxygenation after injury [5]. In animal models of TBI, post-injury hypoxia is associated with inflammation [6,7,8], worse axonal injury [7,8,9], blood–brain barrier disruption [6,10], and functional impairments [6,8,11,12].

We previously reported that after a closed-head weight-drop injury, adolescent mice were at increased risk of death compared to their adult counterparts, with accompanying worse cognitive performance and hippocampal neuronal loss [13]. During these studies, we noted that the increased mortality rate in adolescent mice was related to prolonged bouts of apneic/agonal breathing as compared to adult mice. We found that mice that recovered from apnea/agonal breathing did not display any significant amount of post-TBI mortality. 

Accordingly, we hypothesized that supplemental oxygen would improve survival, potentially by reducing the duration of agonal breathing immediately after injury. However, we were also concerned about the potential side effects of excess oxygen or oxygen toxicity due to the nature of our TBI model, which predominantly induces injury to the optic nerve (i.e., traumatic optic neuropathy (TON). For example, premature infants given supplemental oxygen to treat chronic lung disease have an increased chance of survival [14], but they may also develop retinal fibroplasia, known as retinopathy of prematurity, which is caused by oxygen toxicity [15,16]. Additionally, some retrospective studies have shown that higher brain oxygenation within the first 24 h after TBI is associated with elevated mortality and worse short-term outcomes [17]. Due to this duality of oxygen treatment, we wondered if brief oxygen exposure, lasting only through the duration of agonal breathing, would be sufficient to improve survival without causing oxygen toxicity. We, thus, hypothesized that a minimal dose of oxygen would increase survival, with no effects on other outcome measures.

## 2. Results

### 2.1. Weight, Morbidity, and Mortality following TBI in Adolescent Mice

All mice were weighed for the first 7 days after injury, and mice euthanized on day 30 were weighed before euthanasia. There were main effects of injury (*p* = 0.009) and day (*p* < 0.001) and a significant interaction (*p* < 0.001). Mice given O_2_ weighed significantly less than sham mice throughout the study (*p* = 0.008), while mice not given O_2_ only weighed significantly less on day 30 (*p* = 0.015; Figure 1A). Both TBI groups were also significantly less active than sham animals up to day 11 (TBI: sham *p* = 0.006; TBI + O_2_: sham, *p* = 0.021), with no improvement in mice given O_2_ (TBI: TBI + O_2_, *p* = 0.54). Analysis revealed main effects of day (*p* = 0.004), injury (*p* = 0.005), and light vs. dark (*p* < 0.001; Figure 2B). 

Oxygen exposure varied among mice (range 1.5–4.25 min, mean = 2.95 min, SD = 41.5 s) but had no effect on duration of agonal breathing (*p* = 0.82; Appendix A), with no correlation between these measures (*p* = 0.89; Appendix A). Oxygen exposure also did not affect righting times (*p* = 0.16; Appendix A), with no correlation between agonal breathing and righting time (Appendix A). Strikingly, mortality was significantly reduced in mice given O_2_, from 50.9% (no O_2_) to 13.3% (*p* = 0.001; Figure 2C). Despite our initial thinking that oxygen would improve/shorten agonal breathing times, labored breathing time was not altered in mice given O_2_ (*p* = 0.82) nor was righting time (*p* = 0.16). The duration of oxygen exposure was also not correlated with righting time (*p* = 0.53). However, we noted that oxygen exposure significantly reduced the likelihood of dying if a mouse had apparent seizures after TBI, from 64.8% to 41.2% (*p* = 0.007; Figure 1D). Post-TBI seizures were assessed subjectively by noting changes in animal behavior, including arching of the spine, convulsions, and/or rapid motor movement (usually of the front and rear paws). Data were square transformed to pass normality and equal variance.

### 2.2. Injured Mice Have a Blunted Optokinetic Response, and Oxygen Shows Partial Rescue Acutely

Within the first 7 days post-TBI, injured mice had a significantly blunted optokinetic response across all spatial frequencies assessed: 0.12 cpd (*p* < 0.001), 0.24 cpd (*p* < 0.001), 0.32 cpd (*p* < 0.001), 0.39 cpd (*p* = 0.008), and 0.46 cpd (*p* = 0.002). Mice given oxygen performed similarly to sham mice (*p* = 0.1), while TBI room air mice were significantly impaired (*p* < 0.001) at 0.12 cpd. At 0.24 cpd, both injured groups had blunted OKRs compared to sham, but TBI + O_2_ mice performed significantly better than injured room air mice (*p* < 0.05). At 0.32, 0.39, and 0.46 cpd, both injured groups had significantly blunted OKRs (see Figure 2A). Although both injured groups were impaired compared to sham mice at the upper threshold of rodent visual acuity (0.46 cpd), it is worth noting that 13 of the 24 sham mice still produced an OKR. However, none of our injured mice produced an OKR, indicating a reduction in overall optokinetic function in addition to visual acuity for mice not given O_2_.

Mice were also tested 3 weeks post-injury, at which time the benefits of oxygen and the overall severity of injury-induced OKR deficits were reduced. Effects of injury were only observed in mice given oxygen at 0.24 cpd (*p* = 0.03) and both TBI room air (*p* = 0.02) and TBI + O_2_ (*p* = 0.005) versus sham mice at 0.39 cpd. Of note, there were fewer responses, even in sham mice compared to week 1 (Figure 2B). At 4 weeks, sham mice outperformed TBI mice at 0.24, 0.39, and 0.46 cpd (*p* = 0.03, *p* = 0.003, and *p* = 0.001, respectively, for room air and *p* = 0.02, *p* = 0.006, and *p* < 0.001, respectively, for O_2_). Animals performed similarly with no significant reduction in OKRs in either injured group at 0.12 or 0.32 cpd (*p* = 0.1 and 0.3, respectively; Figure 2C).

### 2.3. Oxygen Slows Acute RGC Loss but Does Not Ultimately Prevent Loss of RGCs

In retinal tissue collected 7 days after injury, cell counts were analyzed in three regions of the retina in order to account for variable distribution of RGCs, as previously described [18]. In the periphery, injury resulted in a significant reduction in Brn3a-positive cells (*p* < 0.001; Figure 3A,D) in both room air (*p* = 0.001) and TBI + O_2_ mice (*p* < 0.001) compared to sham. The same was true in the mid-peripheral retina, with a main effect of injury (*p* = 0.01; Figure 4B) in both room air (*p* = 0.04) and TBI + O_2_ mice (*p* = 0.04) compared to sham. In the center, however, only room air mice had significantly fewer RGCs than sham (*p* < 0.001), while mice given O_2_ retained significantly more cells than their injured counterparts (*p* = 0.01) and had similar numbers to sham (*p* = 0.4; Figure 3C,E).

By thirty days, we found a similar reduction in Brn3a-positive cells in injured groups, room air (*p* = 0.008) and TBI + O_2_ (*p* < 0.001), compared to sham in the periphery (Figure 3F). Similar effects were seen in the mid-periphery with a main effect of injury in room air (*p* < 0.001) and TBI + O_2_ (*p* < 0.001) versus sham (Figure 3G). Interestingly, there were no differences between groups in the central retina (*p* = 0.5; Figure 3H). 

### 2.4. Oxygen Exposure Does Not Prevent Degeneration but May Accelerate Recovery

We examined retinal cell projections in the brain for evidence of axonal degeneration using FJ-C and compared tissue from days 7 and 30 in order to assess the acute and long-term effects of O_2_ after TBI. Beginning in the optic tract (OT), we found a main effect of injury (*p* < 0.001) and a significant interaction (*p* < 0.001, Figure 4). Post-hoc Tukey’s tests showed significant differences between sham and TBI room air (*p* < 0.001), sham, and TBI + O_2_ (*p* < 0.001), and TBI room air versus TBI + O_2_ (*p* < 0.001) at day 7, with O_2_ mice having the highest amount of FJ-C staining. By day 30, sham was still significantly lower than TBI room air (*p* < 0.001) and TBI + O_2_ (*p* < 0.001), but there were no differences between injury groups. There was, however, a significant increase in the FJ-positive area in room air mice from 7 to 30 days (*p* < 0.001), while TBI + O_2_ mice had reduced levels by 30 days (*p* < 0.001). Other regions associated with vision, including the lateral geniculate nucleus and superior colliculi, were also examined, and data are presented in the Appendix A associated with this manuscript (Appendix A).

### 2.5. Astroglial Reactivity Is Significantly Reduced in the Brain following O_2_ Exposure

To determine whether oxygen exposure affected astrocytic responses, we stained brain tissue for GFAP and examined retinal projection regions for increased mean fluorescence intensity (MFI) of GFAP staining. In the optic tract, the pattern of increased fluorescence paralleled the axonal degeneration found with FJ-C. At day 7, there was a main effect of injury (*p* < 0.001), such that both injury groups showed significantly higher GFAP immunofluorescence than their sham counterparts (*p* < 0.001), and TBI + O_2_ mice had significantly lower GFAP expression than room air mice. By day 30, the injury effect (*p* = 0.03) was driven only by the difference between room air and sham mice (*p* = 0.03), while TBI + O_2_ mice were not different than their sham counterparts (*p* = 0.2; Figure 5). Data for additional regions of interest are provided in Appendix A. We also examined microglia morphological changes but found no effects of oxygen nor any differing results from our previous publications, so data were not included in this manuscript.

### 2.6. Oxygen Exposure Promotes IRE-1α Pathway Activation Acutely

Retinas taken on day seven after TBI showed only significant increases in Inositol Requiring Enzyme 1α (IRE1α) pathway activation in mice given oxygen (*p* = 0.04) compared to sham and room-air-exposed mice. There was no detectible increase in the downstream X-box Binding Protein 1 (XBP1-U), its splice variant XBP1s, or the ratio of the two within subjects (Figure 6A–D). By day 30, however, more differences arose in the IRE1α arm of the ER stress response. IRE1α was significantly increased (*p* = 0.004), largely driven by injured room air mice (*p* = 0.003) compared to uninjured mice (Figure 6E). Total XBP1 expression was not altered, but XBP1s showed significant increases in both room air (*p* = 0.01) and TBI + O_2_ mice (*p* = 0.009) compared to sham animals. In addition, the ratio of total to spliced XBP1 showed significantly higher levels of the spliced form in room air (*p* = 0.04) and TBI + O_2_ mice (*p* = 0.002; Figure 6G).

### 2.7. The PERK Pathway Is Sub-Acutely Elevated after TBI, but Supplemental Oxygen Reduces Long-Term Expression of Pro-Apoptotic Markers

Seven days post-injury, the activators associated with the Protein Kinase R-like ER Kinase (PERK) branch of ER stress were similar across groups, with no increases in PERK, phosphorylated (p)-PERK, or the ratio of the two nor in eukaryotic translation initiation factor (eIF2α), p-eIF2α, or its ratio (Figure 7A). Yet, subsequent downstream ER stress factors were elevated in injured mice. Although total activating transcription factor 4 (ATF4) remained stable (*p* = 0.08), there was a significant increase in p-ATF4 (*p* = 0.007) between sham vs. room air (*p* = 0.02) and room air vs. TBI + O_2_ (*p* = 0.01). Upon further analysis, this overall increase in phosphorylation in room air mice was due to proportional levels of total ATF4 (ATF4 ratio vs. sham *p* = 0.25), while TBI + O_2_ mouse retinas contained significantly higher phosphorylated levels vs. sham (*p* = 0.005; Figure 7B). In line with this finding, the ER-stress-associated apoptotic factor C/EBP homologous protein (CHOP) was only significantly increased in mice given oxygen compared to both sham (*p* = 0.005) and injured room air counterparts (*p* = 0.003).

Thirty days post-injury, PERK pathway activation remained elevated in room air mice, but mice given oxygen begin to show adaptive responses (Figure 8). Although total PERK levels are increased in both injured groups (TBI room air *p* = 0.01, TBI + O_2_
*p* = 0.01 vs. sham), p-PERK was unaltered and the ratio of the two was unaffected by oxygen manipulations. Yet, the downstream kinase eIF2α had elevated total protein (*p* = 0.002) and phosphorylated protein (*p* < 0.001). Post hoc analyses revealed that this was driven by increases in TBI + O_2_ mice (*p* = 0.005 and *p* < 0.001, respectively). This increase not only differentiated them from sham animals but also from room air mice (*p* = 0.003 and *p* = 0.001, respectively). At this time point, the downstream transcription factor ATF4 was no longer elevated in either injured cohort, but CHOP remained elevated (*p* = 0.008) in room air mice compared to sham animals (*p* = 0.01) and TBI + O_2_ mice (*p* = 0.01). Post hoc probing of retinal tissue for the PERK arm’s feedback machinery, growth arrest, and DNA damage-inducible protein 34 (GADD34) did not reveal any significant findings (F_2,26_ = 1.8 *p* = 0.18, Appendix A).

We also asked whether oxygen might have an effect on the redox-sensitive protein folding machinery of the ER. ERO1Lα levels were unaltered at day 7 (*p* = 0.52; Figure 9A,C), but oxygen appears to have a delayed effect as far out from injury as 30 days (*p* = 0.03); ERO1Lα levels are reduced compared to sham mice (*p* = 0.03; Figure 9B,D).

## 3. Discussion

We previously found that briefly exposing adolescent mice to oxygen after TBI reduced mortality by 40% compared to adult mice [13]. We now show that oxygen partially rescues optokinetic dysfunction acutely, and these early beneficial effects seem to delay the harmful effects of TBI as time goes on. Moreover, astrocytes showed decreased GFAP expression at both 7 and 30 days in mice given oxygen, with a reduction in some regions comparable to uninjured mice. We show a potential mechanism through which oxygen exerts these protective effects via activation of adaptive ER stress responses (e.g., XBP1s) and later suppression of apoptotic factors (i.e., CHOP).

### 3.1. Supplemental Oxygen Reduces Mortality

Apnea or agonal breathing is associated with a closed-head injury [19]. Interestingly, deaths from apnea are rarely reported in mild–moderate injury paradigms, and only a handful of studies mention the use of oxygen to counteract this injury phenotype [20,21,22]. Despite the occasional acknowledgement of brief oxygen exposure to help mice survive, few studies have explored the effects of supplemental oxygen [23]. We found only three reports where mortality rates paralleled our own and also specified the use of oxygen to improve survival [21,22,24]. Importantly, none of these studies explored how brief oxygen exposure alone might alter injury outcomes. In our previous studies, we observed that mice with prolonged apnea were less likely to recover from the initial impact injury. We found that increasing inhaled oxygen content during this immediate post-injury period was sufficient to keep mice alive until the apneic period ended. In the current study, we examined other potential consequences of keeping these mice alive who might have otherwise not survived injury. Despite the noted survival increase, there were no correlations between agonal (i.e., labored/shallow) breathing and the time spent in the oxygen chamber, nor did O_2_ reduce righting times, suggesting that supplemental oxygen, while improving survivability, did not change the recovery of breathing patterns after TBI.

It is important to note that we did not control the exact duration or dosage of oxygen in this study, and a few studies show that varying the oxygen dosage can alter outcomes. For example, it was shown that in neonates, there was a decreased risk of mortality when given only 21% oxygen versus giving 100% oxygen during neonatal resuscitation [25]. Furthermore, a study in rats observed that a higher saturation of oxygen (91–95%) seems to be safer than targeting a lower saturation of oxygen (85–89%) with respect to blood cell oxidative stress measures [26]. Future studies should explore varying oxygen exposure taking these factors into consideration. Importantly, these studies should note that too lengthy an exposure to 100% oxygen is associated with oxygen toxicity, [27] particularly in the retina [16], while a smaller dose of oxygen (i.e., 50–70%) is associated with more beneficial effects without further damage to visual acuity [27].

### 3.2. Optokinetic Response Outcomes

In addition to the survival benefit, we found behavioral evidence that brief oxygen exposure improved visual performance. We examined the involuntary visually evoked optokinetic nystagmus response, which is blunted in our model, likely due to optic nerve degeneration and retinal ganglion cell loss observed post-injury [13]. Optokinetic nystagmus presents as an easily distinguishable movement of the head in line with a rotating drum holding a visual grating. Injured mice have a blunted OKR at all time points examined and for most spatial frequencies, which were additionally used to assess visual acuity. Mice given supplemental oxygen, though, perform similarly to sham mice acutely (assessed during the first week after injury). This acute protection prevented a significant decline in responses, indicating that oxygen may have slowed the loss of retina-to-brain connections, allowing the OKR to remain functional. Despite this, there was no improvement/retention of visual acuity (shown by zero responses recorded) in both inured groups compared to shams at 0.46 cpd. This distinction becomes relevant when dissecting injury outcomes because acuity can also be affected by a loss of photoreceptors [28,29]. We did not assess changes in photoreceptors in this study, but future studies should consider whether the loss of RGCs is associated with photoreceptor dysfunction in addition to overt RGC loss. 

By 30 days after injury, both injured groups had blunted OKRs and reduced visual acuity, and there was no longer a significant difference between mice given supplemental oxygen vs. room air only. These results could potentially be explained by RGC cell counts—Brn3a-positive cells in the central retina are more numerous in O_2_-exposed mice than in room air mice at day 7. By 30 DPI, this pattern had changed, with the three groups being indistinguishable in central retinal cell counts. This finding of similar RGC counts in the central retina at 30 DPI could be due to a natural decline in RGCs as mice age [30], the prevalence of blood vessels in this region that can limit the number of countable cells, or the fact that Brn3a does not label all RGCs. Future studies could use additional RGC markers to determine whether some cell types are impervious to this decline (e.g., melanopsin-expressing RGCs [31]).

### 3.3. Histology 

FJ-C histology suggests that optic nerve axonal degeneration is higher 7 days after injury in mice given oxygen. However, how could there be increased axonal degeneration with more surviving RGCs? Previous micro-CT data [32] coupled with a lack of positive amyloid precursor staining in the brains of our mice [33] suggest that the location of axon injury in our model is likely inside, or near, the intracanicular portion of the optic canal. This means that the damage to the axons is occurring between the RGC cell bodies in the eye and their projection targets in the brain. Thus, these data are consistent with recent studies distinguishing the distal Wallerian response to axon injury from proximal cell signaling [34]. Wallerian degeneration (WD) is a mechanism of distal axon degeneration that occurs when a damaged segment of axon is severed and cleared after injury, but it describes little as to why this might cause a cell to survive or become apoptotic. Some cells can survive injury if only a branch of the axon is damaged [35]; these cells release fewer death-associated factors after injury, potentially saving cells despite loss of their distal portion. WD also involves several redox-associated mechanisms, which might be sensitive to alterations in oxygen delivery, thus leading to the worse distal axon degeneration that we see acutely in room air mice. 

Moreover, due to the slow nature of WD in the central nervous system, there may be less potential for axon regeneration without intervention, as compared to the peripheral nervous system. WD mechanisms are tightly linked with the redox state of the axon/injured cell [36], and a mutant mouse with slowed WD is purported to have increased neuroprotection from axon injury due to increased NAD+ activation and, thus, to decreased oxidative stress [37,38].

Hypoxia, cerebral edema, and increased intracranial/intraocular pressure all induce oxidative stress [37]. Although our model of TBI is not a hypoxia model, and we did not measure intraocular pressure, the duration of apnea, compression of the orbital bone over the optic nerve, and likely decreased oxygen supply to the brain for a brief period of time could lead to toxic oxidative reactions. Therefore, it is reasonable to ask whether this brief oxygen intervention alters some of the WD mechanisms of axon degeneration to speed up the degeneration of those cells more likely to die (i.e., leading to a fast peak degeneration at day 7) but promoting survival in those that were “savable” (i.e., leading to decreased degeneration by day 30). Future research will need to determine the role of oxidative stress in this model and whether brief oxygen exposure is sufficient to alter this state in the axons.

Another explanation for our results could be that the time points examined do not represent peaks and troughs of degeneration in each group. It is possible that degeneration in the room air mice is progressing at a slower rate, peaking some time before, or around, day 30, where we see higher amounts of FJ-B staining compared to injured O_2_ mice. Conversely, day 7 may represent the peak of degeneration for O_2_ mice, which could have been accelerated and then diminished by day 30. It is also possible that cell death was not truly affected, and the reduction in FJ-B in oxygen mice is simply the result of improved glial-mediated clearance of axonal debris or further slowing of WD processes. Indeed, we and others have discussed the potentially dysfunctional role of glia after TBI, and we have shown that reactive glia fail to clear axonal debris chronically throughout the visual system [39]. Future research will need to examine tissue at later time points and consider interactions with other cell types to determine whether this reduction given oxygen is associated with more efficient phagocytosis in glial cells. Future research will also need to trace which axons may be resilient to this injury and whether they do eventually die off due to aberrant secondary injury cascades. 

Though our intervention is not a hyperbaric chamber, hyperbaric oxygen therapy (HBOT) reduces neuronal loss and suppresses microglial inflammatory responses but aggravates astrocyte activation in an animal model of hypoxia [40]. Conversely, in pain models using HBOT, there is a reduction of astroglial activation [41,42]. This disease-specific/region-specific response of astrocytes to changing redox environments suggests that a difference in injury or oxygen concentration might lead to unique cellular responses. In particular, we were interested in astrocytes because they are attuned to decreases in oxygen content as small as a few mm of Hg [43] and have been shown to increase activity under oxygen deprivation [7]. In this study, astrocyte reactivity was significantly decreased in mice given oxygen compared to sham animals in nearly all optic-associated brain regions examined at 7 days post-injury. Oxygen could have served less as a direct intervention and more as preconditioning, as astrocytes pre-exposed to hyperoxia express more glutamate transporters, poising them to more readily clear excess glutamate from synapses, a common event after TBI [44]. 

It is important to note that oxygen did not prevent reactive astrogliosis, as TBI + O_2_ mice still had significantly higher expression in the OT at 7 days. However, by thirty days, mice given oxygen had similar levels of GFAP to control mice. This pattern might be partially explained by our FJ-C data. Glial cells are a necessary response to Wallerian degeneration and axon injury response [36], so their early presence and subsequent decline could simply be due to corresponding decreases in degenerative debris. It is also possible that these changes in reactivity could represent shifts between different astrocytic functional states. TBI-induced reactivity in astrocytes can be beneficial (e.g., tissue repair and synaptic remodeling) or detrimental (e.g., impaired neuronal signaling or increase pro-inflammatory cytokines), and these responses are attuned to an astrocyte’s particular microenvironment, allowing them to change based on the degree of axonal degeneration [45]. Much more research into the mechanistic role of oxygen exposure in the eye and brain in relation to both RGCs and glia is needed to answer these questions, though. 

### 3.4. ER Stress

Because of the relationship between hypoxia and oxidative stress, we examined a cellular stress-response mechanism that is sensitive to changes in the oxidative environment and that is elevated after TON [13]. ER stress results in a dynamic range of transcriptional and translational signals in response to any disturbance in the protein folding capabilities of the ER. This response is activated when any of three associated receptors—inositol-requiring enzyme 1α (IRE1α), PRK-like Endoplasmic Reticulum Kinase (PERK), or activating transcription factor 6 (ATF6)—senses an abundance of misfolded proteins in the ER lumen. Each of these receptors can activate a signaling cascade to make acute corrections, prolong adaptive change, or induce apoptosis when the burden is too great [46] Targeting ER stress responses after TBI is predominantly associated with reduced neuron loss/apoptosis, most often in connection with the suppression of IRE1α and PERK branches [47,48,49,50,51,52,53]. Adaptive ER stress responses include the shunting of protein translation through phosphorylation of eIF2α, translation of ATF4, and activation of mRNA splicing XBP1s. Each of these proteins reduces the burden on the ER by preventing new protein translation, upregulation of antioxidant and biogenic protein translation, or destruction of premature mRNA, but their roles in chronic activation are unknown [46]. ER stress-mediated apoptosis, on the other hand, typically subsides in the acute phase, as cells that are too damaged are destroyed.

Therefore, much of what we know about ER stress in living organisms focuses on our understanding of acute adaptive responses and apoptosis, leaving a gap in understanding the effects of long-term activation. Our results suggest that ER stress is likely involved in RGC cell fate mechanisms at least 30 days post-injury. Injured mice showed increased PERK pathway activation via upregulated ATF4 translation (i.e., total ATF4) but not phosphorylation. This distinction is important because increases in ATF4 translation can be both adaptive (e.g., induction of antioxidant defense) or apoptotic (i.e., activation of CHOP) and because phosphorylation of ATF4 is associated with its degradation [54,55] and the termination of ER stress activation. Thus, at day 7, ER stress continues in injured mice but may be declining in mice given oxygen. We attempted to further test this hypothesis by measuring PERK pathway feedback using GADD34, which dephosphorylates eIF2α after ER stress subsides. These data were not significant (Appendix A), but examination of other time points may prove informative. 

Although the pro-apoptotic CHOP is still higher in O_2_ mice than in controls at 7 days, an adaptive (rather than apoptotic) effect of oxygen is supported by the normalization of CHOP expression at 30 days after injury, as compared to room air mice. Further support for an adaptive effect of oxygen treatment comes from the early upregulation of the IRE1α pathway, which is associated with attempts to decrease/protect against prolonged ER stress [56]. With room air only, this pathway’s response may be delayed, as elevated CHOP expression is not seen until 30 days. It will be important for future studies to determine more about this flux in ER stress pathway responses across time since retinal cells appear to remain stressed, even though apoptosis mostly subsides between 7 and 30 days based on Brn3a cell counts. 

A related cellular stress pathway relevant to these studies is oxidative stress. In studies assessing the effects of brief, acute oxygen exposure, toxicity is correlated with both the concentration of oxygen delivered [26] and the strain imposed on the pulmonary system (e.g., in patients with chronic obstructive pulmonary disease) during exposure [57]. A more taxed pulmonary system, when briefly exposed to oxygen, produces fewer reactive oxygen species than when no oxygen is supplied, significantly reducing free radicals in the lungs [57]. We may, thus, be able to ask whether brief oxygen exposure is reducing the increased expression of reactive oxygen species typically associated with TBI [58,59,60,61].

If this is the case, reduced ROS could explain the shift to an adaptive ER over an apoptotic one, due to the overlap of these two systems [52,62,63,64,65,66,67,68,69,70,71,72,73,74,75]. Although not a direct measure of free radicals or antioxidant response, we examined protein expression for the highly redox-regulated protein folding machinery of the ER, ERO1Lα. Oxygen significantly reduces ERO1Lα, even compared to control mice. In cases of antioxidant interventions, ERO1Lα reduction is related to reduced free-radical formation and a reduction in ER burden [76,77]. Although little research has been conducted in disease models to explain the interactions between ER stress and oxidative stress, the two pathways overlap heavily with both upstream and downstream effectors (e.g., integrated stress responses to hypoxia, mitochondrial associations, PERK, and the JNK pathway) that could reasonably control either mechanism (e.g. [75,76,77,78,79,80]). Given the acute and chronic effects of this minor reduction in oxidative burden on the CNS in our mice given oxygen, future studies will need to consider the role of ROS and begin unraveling the mechanisms of ER stress and oxidative stress in the context of TBI/axon injury. 

## 4. Materials and Methods

### 4.1. Animals

Experiments were performed in 6-week-old adolescent male C57BL/6J mice (Jackson Laboratories, Bar Harbor, ME, USA). Mice were housed under a 14 h:10 h light/dark schedule in pressurized individually ventilated cage racks, with 4 mice per cage, and were given ad libitum access to water and standard rodent chow. Animals habituated to the vivarium for one week prior to undergoing traumatic brain injury and subsequent procedures. The University of Cincinnati Institutional Animal Care and Use Committee approved all experimental procedures under protocol 20-02-26-02. 

### 4.2. Traumatic Brain Injury/Traumatic Optic Neuropathy

Closed-head injury was performed by weight drop, as previously described [32,81]. [Briefly, mice were anesthetized using isoflurane (2–3%) and placed in prone position under a metal rod raised above the intact, unshaven scalp. The rod was dropped 1.5 cm, roughly above bregma (Figure 10A,B). After head trauma, mice were placed under an oxygen hood made from a pipet box connected to an oxygen tank and were exposed to 100% oxygen (O_2_) until normal breathing returned—approximately 1–5 min (Figure 10B)—or were allowed to recover in room air. Mice were observed for recovery of righting reflex before being returned to their home cages. Injured mice not given oxygen will hereon be referred to as room air Mice, while sham mice will be referred to as sham. Sham animals were anesthetized, weighed, and allowed to recover before being returned to their cages. We performed this protocol in three cohorts of mice for three behavioral time points and various molecular and histological measures described below.

### 4.3. Behavior

#### 4.3.1. Home Cage Activity Monitoring

Twenty-four hours post-injury, animals were weighed and separated into individual cages with food and water but without enrichment (e.g., no cotton bedding). Cages were situated in activity monitors (San Diego Instruments PAS system, N = 12; and Lafayette Activity Monitoring System, N = 8) set to record movements for 24–48-hour time blocks. Every 24–48 h, mice were weighed, and the systems were reset. This was repeated for 15 days post-injury (cohort 2, Figure 10D). Monitors malfunctioned between days 3 and 6 after a weekend power outage that reset the system, so measurements are only present for days 1–2 and 7–15. The number of beam breaks was totaled during each light or dark phase on each day.

#### 4.3.2. Optokinetic Response (OKR) and Visual Acuity

As described in Hetzer et al. [18], we measured optokinetic responses and visual acuity. Briefly, mice were placed on an immobile platform surrounded by a rotating Plexiglas drum. An exchangeable series of sine-wave gratings were placed around the inside of the drum. For these experiments, five gratings were used (0.12, 0.26, 0.32, and 0.49 cycles per degree). Once in the machine, mice were left for a one-minute habituation period before the machine was turned on. The drum rotates at two revolutions per minute (rpm) for two minutes in either clockwise or counterclockwise rotation, followed by 30 s rest, and two minutes rotation in the opposite direction. Mice were exposed to one grating per day in random orders. Experimenters were blinded to conditions during both behavior and scoring. Videos were scored by trained experimenters and the number of optokinetic responses tallied. Two scorers reviewed each video.

### 4.4. Histology

#### 4.4.1. Tissue Collection

For immunohistochemical and immunofluorescence (IHC/IF) analyses, mice were euthanized using Fatal Plus^®^ on days 7 or 30 after TBI. Mice were perfused transcardially with 4% paraformaldehyde in 0.02M phosphate-buffered saline (PBS) solution (pH 7.4). Brains were removed and post-fixed in 4% paraformaldehyde in 0.02M PBS for 24 h, rinsed in 0.01M PBS, and immersed in 30% sucrose solution at 4 °C until sectioning. Sucrose-saturated brains were frozen on dry ice and sectioned at 30 μm using a sliding microtome (Leica, Bannockburn, IL, USA). Sections were stored in cryoprotectant solution (0.01M PBS, polyvinyl-pyrrolidone (Sigma-Aldrich, Burlington, MA, USA; Cat# PVP-40), ethylene glycol (Fisher Scientific, Waltham, MA, USA; Cat # E178-4), and sucrose (Fisher Scientific; Cat # S5-3)) at −20 °C until staining was performed. The left eye was also removed after perfusion, post-fixed in 4% PFA for 4 h, switched to 30% sucrose, and stored at 4 °C until histological staining was performed (see Hetzer, et al., 2021) [18].

For Western blotting, the right eye was taken from mice after injection with Fatal Plus^®^ but before perfusion. After removal, the eye was placed in ice-cold 0.01M PBS, and the retina was removed within 5 min, immersed in cold lysis buffer (described below), and frozen on dry ice. Retinas were stored at −80 °C until used for Western blotting.

#### 4.4.2. FluoroJade-C

Fluoro-Jade C (FJ-C; Histo-Chem, Jackson, AR; Cat# 1FJC), a marker for degenerating neurons and axons [82], was used to stain tissue sections according to the manufacturer’s directions, with slight modifications to avoid high background. Sections were incubated in 0.06% potassium permanganate for 5 min and then submerged in 0.0001% FJ-C solution for 5 min. After staining, slides air-dried in the dark. Slides were stored without coverslips in a slide box and imaged immediately to reduce background quenching of positive signals. 

#### 4.4.3. Immunofluorescence

Free-floating brain tissue was stained for polyclonal rabbit anti-glial fibrillary acidic protein primary antibody (GFAP; DAKO, Santa Clara, CA, USA; Cat # Z0334; RRID AB_10013382) overnight at 4 °C at 1:2000 dilution. On the second day, tissue was incubated with Cy-3 conjugated secondary antibody (Jackson Immunochemicals, West Grove, PA; Cat# 711-165-152, RRID AB_2307443) at 1:500 dilution for 1 h at room temperature. Slides were cover slipped using the antifading polyvinyl alcohol mounting medium gelvatol (Sigma-Aldrich, St. Louis, MO, USA). Increased GFAP immunoreactivity (i.e., mean florescence intensity; MFI) was measured as an indicator of astrogliosis. 

Whole retinas were flat-mounted on slides coated with Gatenby’s solution and stained sequentially using the following primary antibodies: brain-specific homeobox/POU domain protein 3A (Brn3a; Millipore; Cat #MAB1585; RRID: AB_94166) at 1:1000, then GFAP (1:1000). DAPI nuclear staining was achieved with Vectashield Antifade Mounting Medium with DAPI (Vector Laboratories; Cat # H-1200; RRID: AB_2336790). After Brn3a incubation, retinas were washed in 0.5% TX-100, incubated in anti-mouse biotinylated secondary antibody (1:400; Vector Laboratories; Cat # BA-9200; RRID: AB_2336171) for 1.5 h, incubated in VECTASTAIN Elite ABC Kit (1:800; Vector Laboratories; Cat # PK-6100; RRID: AB_ 2336817) for 1 h, then incubated in Cy3 streptavidin (1:500; Invitrogen, Grand Island, NY; Cat # 434315) for 2 h at room temperature covered. This was followed by GFAP overnight incubation and subsequent labeling with anti-rabbit Alexa 488 (1:500; Invitrogen; Cat # A11034). GFAP staining in retinal flat mounts was inconsistent within and between conditions and, thus, was not analyzed. 

### 4.5. Image Analysis

Image analysis was performed as described previously [18]. Briefly, 10× and 20× images of left and right regions of interest (ROIs) for GFAP slides were photographed using an Axio lmager Z1 microscope with an Apotome (Leica Microsystems, Buffalo Grove, IL, USA). For measurement of GFAP MFI, all slides were photographed using the same exposure and magnification between treatment conditions. FJ-C brain slides and retinal whole mounts were imaged on a Nikon C2 Plus Confocal Microscope (Nikon Corporation, Melville, NY, USA). A blinded observer took all pictures. Image analysis and quantification of GFAP MFI were also performed by a blinded investigator using ImageJ software version 1.53t [83] across four non-overlapping 150 × 150 pixel square sections for both left and right ROIs. FJ-C-positive area and Brn3a cell counts were measured using Nikon Elements Analysis software (version 4.6; Nikon, Melville, NY, USA).

### 4.6. Western Blots

Right retinas were homogenized in freshly prepared lysis buffer (20 mM Tris-HCL pH 7.4, 2 mM EDTA, 0.5 mM EGTA, 1 mM DTT, HALT protease/phosphatase inhibitor) using a pellet homogenizer, centrifuged at 3000 rpm for 20 min, and supernatant removed for protein concentration analysis using a BCA protein assay (Pierce BCA Protein Assay Kit; Thermo Fisher Scientific; Cat # 23227). Thus, 20 or 30 µg samples were loaded into SDS-PAGE gels and transferred onto either Amersham Hybond-P 0.45 µm PVDF membranes (GE Life Sciences, Pittsburgh, PA, USA; Cat# GE 10600029) or 0.45 µm nitrocellulose membranes (Bio-Rad, Hercules, CA, USA; Cat# 1620145). Membranes were incubated in Fisher No-Stain Total Protein Stain (Thermo Fisher, Cat# A44449) as per manufacturer instructions, blocked in either 5% non-fat milk or 5% BSA (manufacturer-dependent) for 1 h at room temperature, followed by overnight incubation in primary antibodies at 4 °C (see antibody Table 1). Membranes were rinsed in TBST then incubated in anti-rabbit HRP for 1.5–2 h at room temperature. Blots were imaged using an iBright™ Imaging System (Thermo Fisher) and analyzed using Image J. Some blots were stripped following imaging in stripping buffer (β-mercaptoethanol, 20% Sodium Dodecyl Sulfate, and 1M Tris-HCl pH 6.8) for 30 min at 50 °C, washed, re-blocked, and exposed to the same immunoblotting steps as above. 

### 4.7. Statistical Analysis

For all analyses, alpha was set a priori to *p* < 0.05. Weight change was analyzed using 2-way ANOVA with repeated measures (injury × day), activity monitoring through 3-way ANOVA with repeated measures (injury × day × light), mortality using student’s *t*-test, and seizure morbidity with X^2^ using SigmaPlot 14 (Systat Software, San Jose, CA, USA). Remaining analyses were performed and results graphed using GraphPad Prism 9 (GraphPad Software, San Diego, CA, USA). For OKR data, one-way ANOVAs were analyzed for each spatial frequency but were combined onto one graph for presentation purposes (e.g., Figure 3). FJ-C data were analyzed via 2-way ANOVA. The background and exposure differences inherent in GFAP fluorescence intensity analyses, however, were only conducive to analysis within time points. One-way ANOVA was also used for all Western blot data, which were first normalized to total protein, and similarly to OKR data, some proteins were combined on the same graphs for cohesion (e.g., Figure 7, Figure 8 and Figure 9). Table 2 shows detailed test statistics arranged according to their respective results sections. Appendix A analyses are described in Appendix A legends.

## 5. Conclusions

In conclusion, supplemental oxygen might be advantageous after TBI in more ways than simply improving the post-TBI apneic period and enabling increased survival. Several mouse models of TBI report the use of/need for supplemental oxygen after TBIs of various severities to promote survival due to this prolonged apnea. However, the potential for other effects of oxygen alone has not yet been thoroughly examined. The current data supports a need to better understand the effects of such oxygen used for post-injury resuscitation. This study also suggests several potential mechanisms through which oxygen may be affecting recovery (Figure 11). These include changes in oxidative and/or endoplasmic reticulum stress responses. Understanding the relationship between oxygen and retinal ER stress responses could potentially make it a target for interventions to improve axonal injury after TBI. 

## Figures and Tables

**Figure 1 ijms-24-09831-f001:**
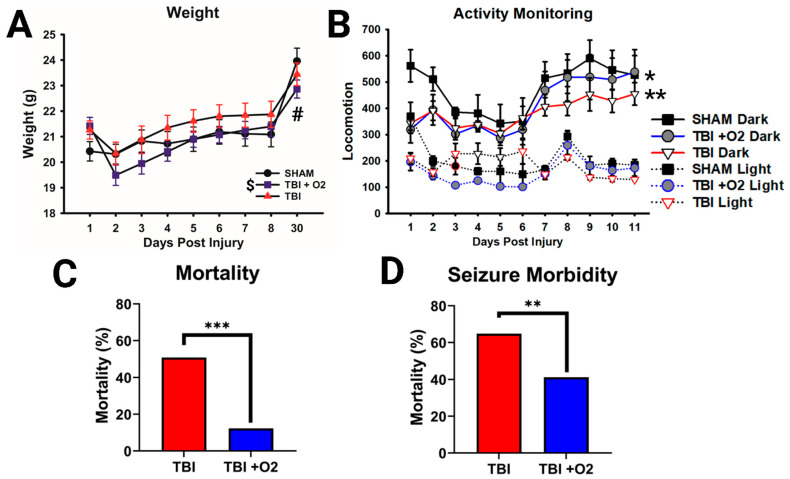
Post-TBI outcomes: mortality and morbidity. (**A**) TBI and sham mice weighed similar amounts until the final time point examined at day 30 (#), while mice given oxygen weighed significantly less than sham mice throughout the study (**). (**B**) Injured mice were also significantly less active than sham mice, particularly during their active dark phase. (**C**) Adolescent mice have a mortality of 50%, but oxygen reduces this to 13% in addition to (**D**) reducing the likelihood of mortality when a tonic-clonic seizure occurs. * *p* < 0.05, ** *p* < 0.01, *** *p* < 0.001, $ = overall TBI +O_2_ vs. sham, # = TBI vs. sham day 11 only.

**Figure 2 ijms-24-09831-f002:**
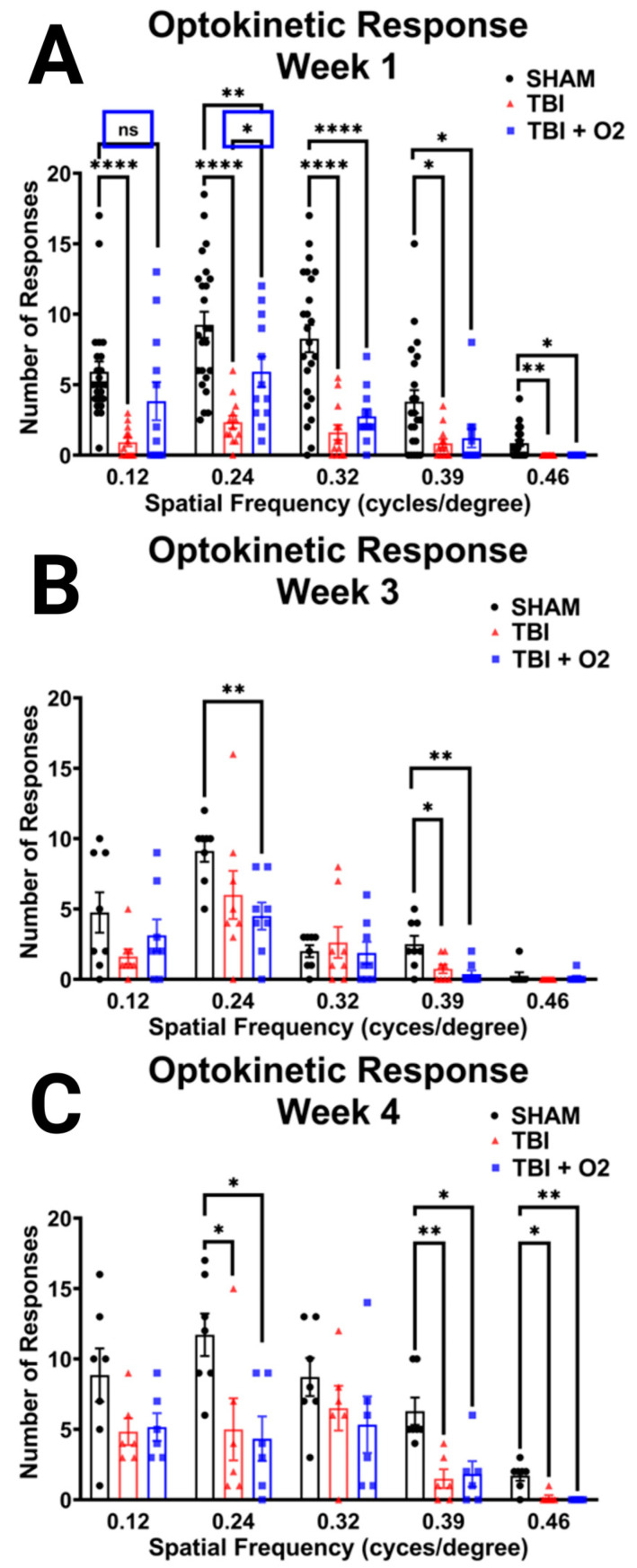
Oxygen modestly rescues acute OKR but does not prevent sub-acute decline. (**A**) OKR was measured in the first week after TBI, revealing significant impairment at each cpd in injured mice. TBI + O_2_ mice performed significantly better than room air counterparts at 0.12 and 0.24 cpd (blue boxes). (**B**) Protective effects were no longer present at 3 weeks post-injury or (**C**) 4 weeks post-injury. * *p* < 0.05, ** *p* < 0.01, **** *p* < 0.0001, ns = not significant.

**Figure 3 ijms-24-09831-f003:**
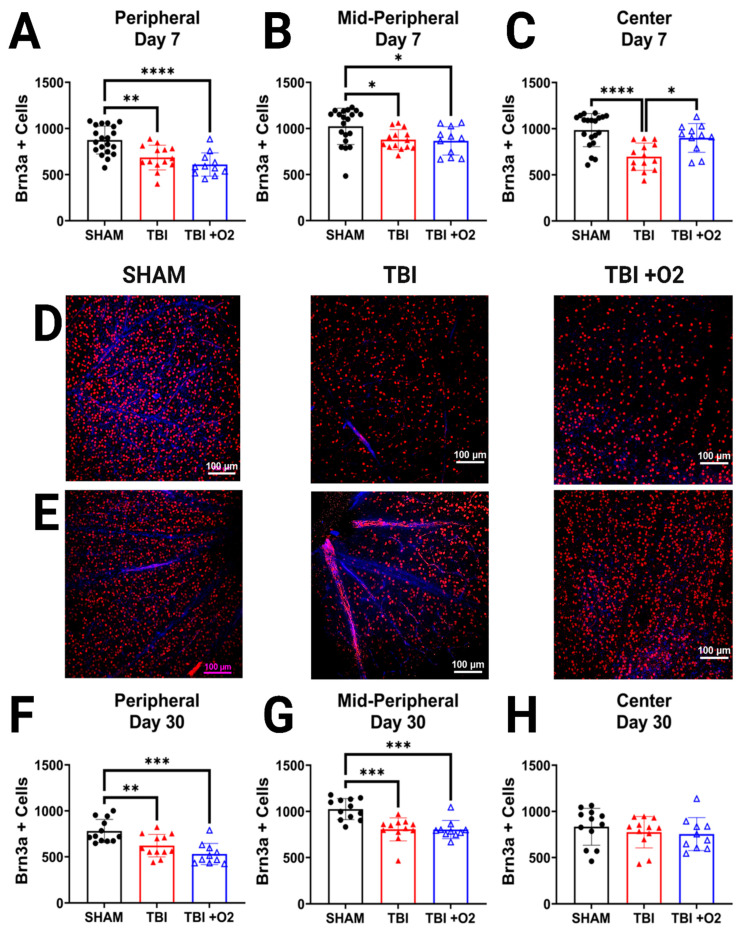
Oxygen prevents acute loss of central retinal cells but not overall loss by 30 days. RGCs were measured in three quadrants of the retina—(**A**) peripheral, (**B**), mid-peripheral, and (**C**) central 7 days post-injury. Representative photomicrographs taken at 20x magnification for (**D**) peripheral RGCs (red) and DAPI (blue) of sham, room air, and TBI + O_2_ mice and (**E**) central RGCs, respectively. From cohorts 2 and 3, 30-day retinas were also stained for Brn3a (red) and analyzed in (**F**) peripheral, (**G**) mid-peripheral, and (**H**) central quadrants. Scale bars represent 100 µm. * *p* < 0.05, ** *p* < 0.01, *** *p* < 0.001, **** *p* < 0.0001.

**Figure 4 ijms-24-09831-f004:**
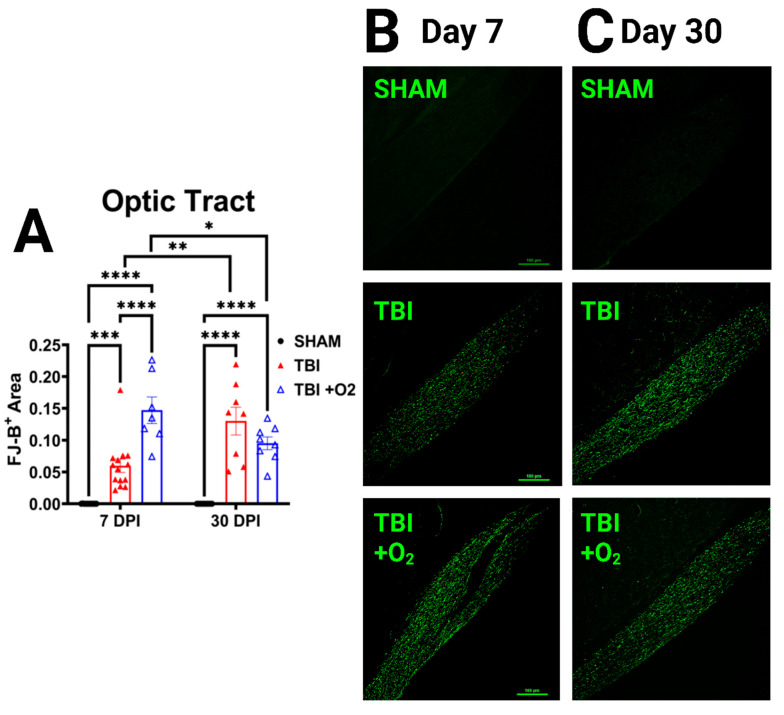
Injured mice given oxygen show increased axonal degeneration acutely. (**A**) Graphic representation of positive FJ-B-stained area in the optic tract. (**B**) Representative photomicrographs of FJ staining in the OT in 7-day tissue for sham, room air, and TBI + O_2_ mice. (**C**) Representative photomicrographs of FJ staining in the OT in 30-day tissue. Scale bars represent 100 µm. * *p* < 0.05, ** *p* < 0.01, *** *p* < 0.001, **** *p* < 0.0001.

**Figure 5 ijms-24-09831-f005:**
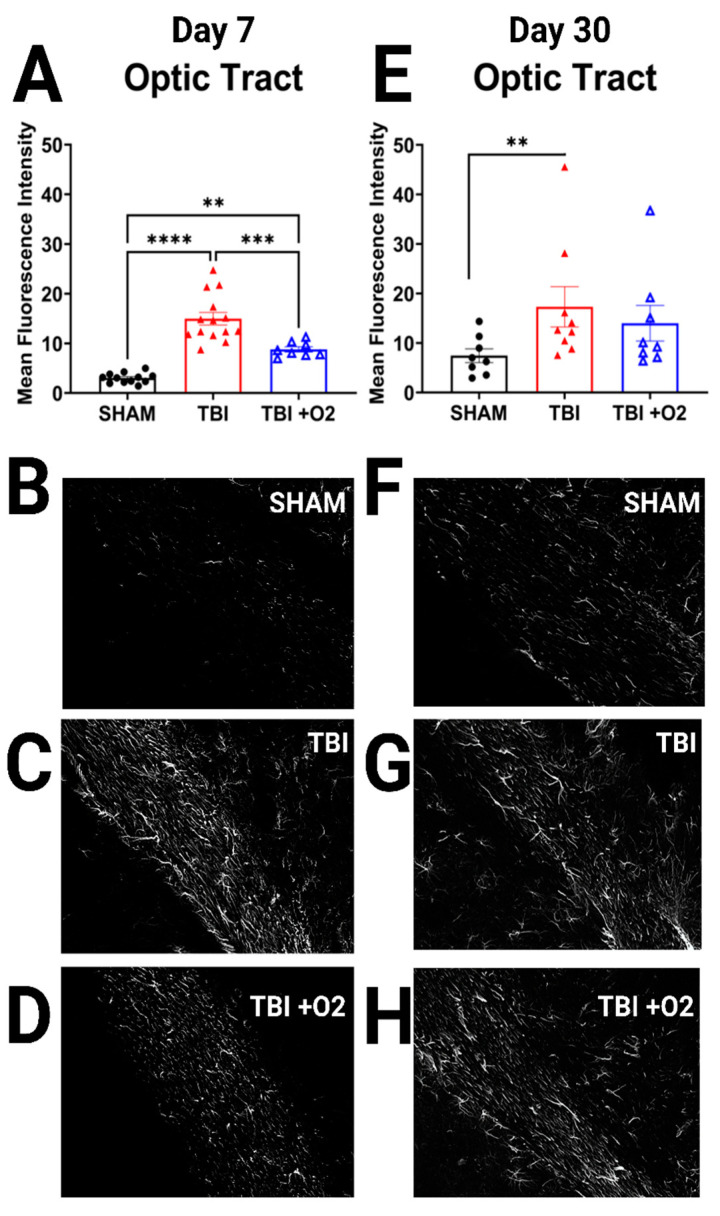
Oxygen decreases astroglial reactivity. (**A**) Measures of mean fluorescence intensity for GFAP staining show increased GFAP immunostaining in both injured groups but significantly less in mice given oxygen compared to those not. (**E**) This effect is paralleled in 30-day tissue where mice given oxygen are not different from uninjured counterparts. (**B**–**D**) Representative images of GFAP in sham, room air, and TBI + O_2_ mice, respectively, at 7 days. (**F**–**H**) Groups at 30 days. Scale bars represent 100 µm. ** *p* < 0.01, *** *p* < 0.001, **** *p* < 0.0001.

**Figure 6 ijms-24-09831-f006:**
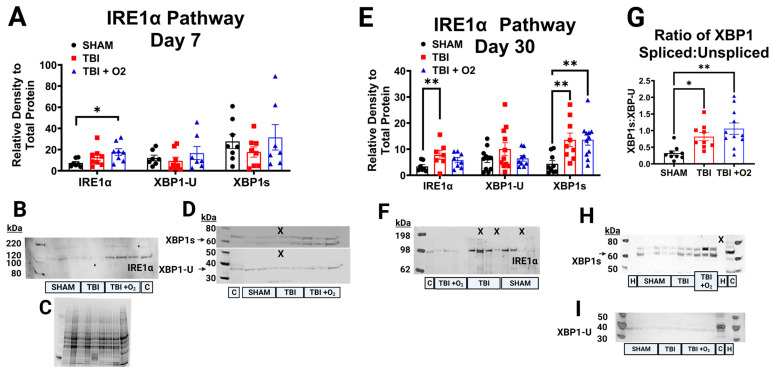
IRE1α Pathway activation after oxygen exposure. (**A**) Relative densities of total IRE1α, XBP1 unspliced, and XBP1 spliced show acute activation only in mice given O_2_ in 7-day tissue. (**B**) A representative blot of IRE1α shows darker bands at the predicted 100 kDa molecular weight in mice given O_2_. (**C**) A total protein blot image, which was used for normalization. (**D**) Representative blots of XBP1s at ~55 kDa and XBP1u at ~38 kDa with their associations. (**E**) Similar data in 30-day retinal tissue now with significant elevation of the IRE1α-ERAD pathway in both inured groups, where not only total expression of XBP1s was found but the (**G**) ratio of spliced to unspliced was significant. Representative blots for IRE1α (**F**), (**H**) XBP1s, and (**I**) XBP1u. C = control lane for intermembrane control brain homogenate, H = water negative control lane, X = lane not measured, kDa = Killa Dalton. * *p* < 0.05, ** *p* < 0.01.

**Figure 7 ijms-24-09831-f007:**
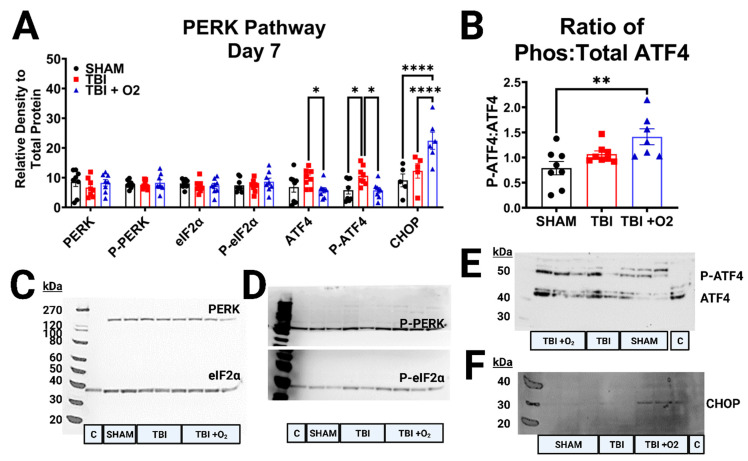
Perk Pathway Activation 7 days post-injury. (**A**) Analyses of PERK pathway markers, PERK, p-PERK, eIF2α, P-eIF2α, ATF4, p-ATF4, and CHOP were measured 7 days post-injury and late-stage PERK pathway activation in the form of elevated antioxidant and apoptotic transcription factors ATF4 and CHOP, respectively, were elevated in injured mice, with higher expression in mice given oxygen for both CHOP and (**B**) the phospho-Atf4. (**C**) Representative blots of PERK (~140 kDa) and eIF2α (38 kDa), (**D**) p-PERK (~110 kDa), and P-eIF2α (38 kDa), and (**E**) ATF4 (Total ~48 kDa, phospho ~60 kDa), and (**F**) CHOP (~30 kDa). C = control lane for intermembrane control brain homogenate, H = water negative control lane, kDa = Killa Dalton. * *p* < 0.05, ** *p* < 0.01, **** *p* < 0.0001.

**Figure 8 ijms-24-09831-f008:**
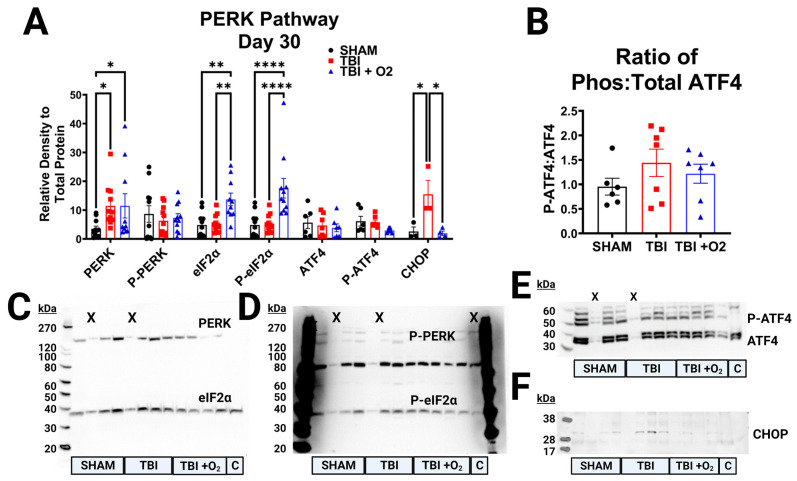
PERK Pathway Activation 30 days post-injury. (**A**) Analyses of PERK pathway markers, PERK, p-PERK, eIF2α, p-eIF2α, ATF4, p-ATF4, and CHOP were measured 30 days post-injury and appear to remain active at this chronic time point, predominantly in room air mice who still express apoptotic CHOP while +O2 mice no longer do despite eIF2α upregulation. (**B**) There was no longer elevated phospho-ATF4 to explain persistent CHOP elevation in room air mice. (**C**) Representative blots of PERK and eIF2α, (**D**) p-PERK, and P-eIF2α, and (**E**) ATF4, and (**F**) CHOP. C = control lane for intermembrane control brain homogenate, X = lane not measured, kDa = Killa Dalton. * *p* < 0.05, ** *p* < 0.01, **** *p* < 0.0001.

**Figure 9 ijms-24-09831-f009:**
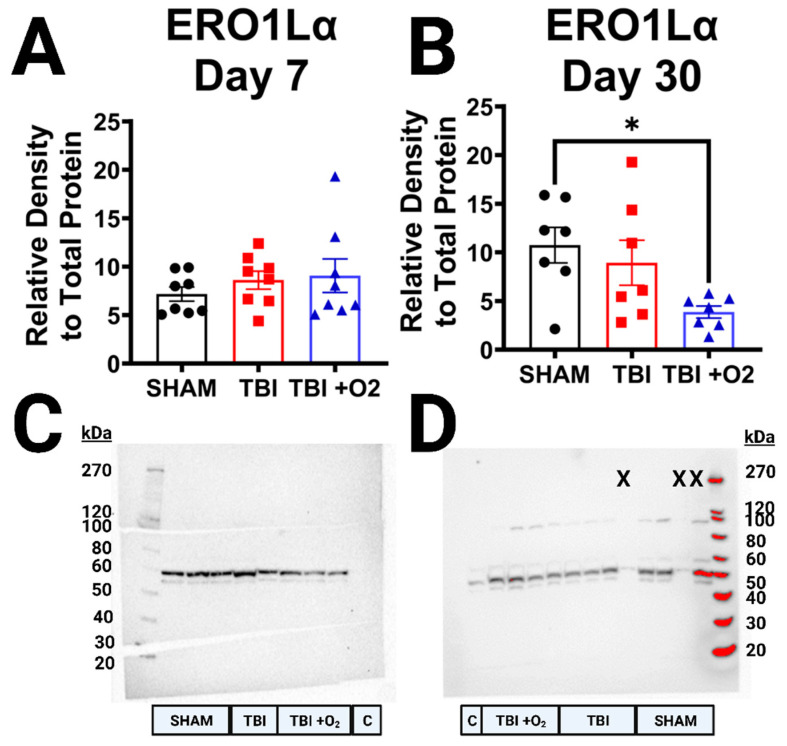
ERO1Lα reduction 30 days post-injury after brief oxygen exposure. (**A**) Analysis of ERO1Lα marker at 7 days post-injury. There are no significant differences between sham, room air mice, and mice given oxygen. (**B**) Mice given oxygen have a significant decrease in oxidoreductase marker ERO1Lα compared to sham mice. (**C**) Representative blot of ERO1Lα at 7 days post-injury. (**D**) Representative blot of ERO1Lα at 30 days post-injury. C = control lane for intermembrane control brain homogenate, X = lane not measured, kDa = Killa Dalton. * *p* < 0.05.

**Figure 10 ijms-24-09831-f010:**
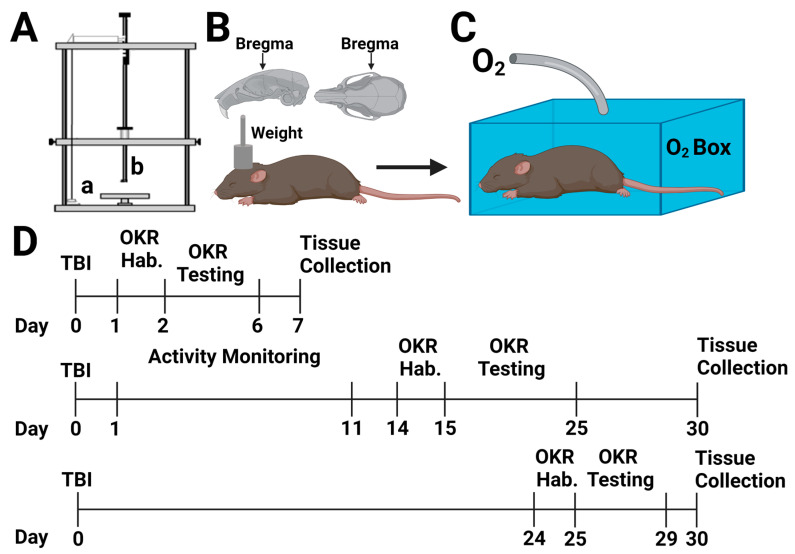
Experimental procedure and timeline. (**A**) For closed-head weight-drop injury, anesthetized mice are placed in the prone position on a cork platform (a) and a 400 g weight (b) is dropped roughly above bregma, as depicted in (**B**). (**C**) Brief oxygen exposure was induced by placing mice in a modified pipette box with a tube delivering 100% oxygen until normal breathing returned. (**D**) Three cohorts of mice were used for these studies with varying optokinetic testing times and tissue collection times as shown with each of the timelines (Traumatic Brain Injury: TBI; OKR: Optokinetic Response; Hab: Habituation). This figure was created using BioRender.

**Figure 11 ijms-24-09831-f011:**
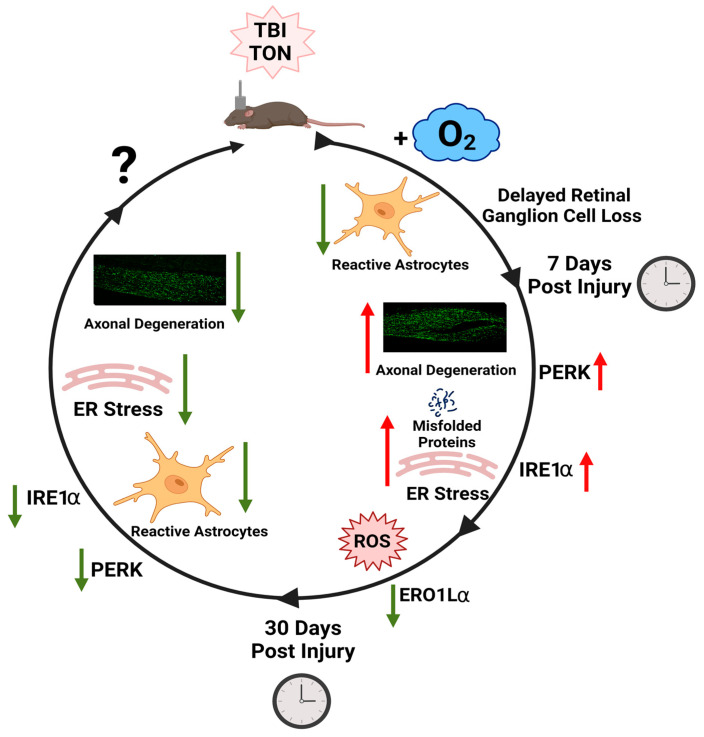
Summary of Findings. We present a potential mechanism of oxygen’s actions in retinal ganglion cells and their axons after TBI-induced TON. Based on our findings, oxygen impacts astrocyte reactivity, RGC survival, and ER stress dynamically over time. We hypothesize that oxygen reduces the redox environment mediated via ERO1Lα, which then communicates to the retinal ER to shift from an apoptotic response to one of adaptation. Because we have not examined outcomes for the last 30 days, and little literature exists on chronic ER stress or the potential for long-term reductions in degeneration, we leave a “?” to signify a need for future research. This figure was created using BioRender.

**Table 1 ijms-24-09831-t001:** Antibodies.

Antibody	Concentration	Host Species	Molecular Weight (kDa)	Supplier	Cat #	RRID	Immunogen
GFAP	1:2000	Rbt	N/A	DAKO (Agilent)	Cat: Z033401-2	AB_10013382	Whole bovine GFAP, isolated from spinal cord
Cy3 AffiniPure Donkey anti Rabbit IgG (H+L) conjugated secondary	1:500	Donkey	N/A	Jackson Immuno Research	Cat: 711-165-152	AB_2307443	Gamma immunoglobulins, heavy and light chains
ATF4 + p-ATF4	1:1500	Rbt Poly	48, 70	Fisher	Cat: 10835-1-AP	AB_2058600	UniProt AG1279
CHOP/GADD153	1:1000	Rbt Poly	23	Novus Bio	Cat: NBP2-13172		UniProt P35638
eIF2α	1:2500	Rbt Poly	38	CST	Cat: 5324	AB_10692650	UniProt P05198
P-eIF2α	1:500	Rbt Poly	38	CST	Cat: 3398	AB_2096481	UniProt P05198
ERO1L α	1:2500	Rbt Mono	50	Fisher	Cat: 702709	AB_2716886	Protein corresponding to human ERO1L (aa22-aa468)
GADD34	1:750	Rbt Poly	73	ProteinTech	Cat: 10449-1-AP	AB_2168724	GADD34 fusion protein Ag0578
IRE1α	1:1000	Rbt Poly	100	Novus Bio	Cat: NB100-2324SS	AB_10000972	Swiss-Prot #O75460
PERK	1:1000	Rbt Poly	140	CST	Cat: 5683	AB_10841299	UniProt Q9NZJ5
XBP1U	1:500	Rbt Poly	38	Fisher	Cat: 25977	AB_2880326	XBP1 Fusion Protein Ag21714 (167-261 aa encoded by BC000938)
XBP1s	1:500	Rbt Poly	45/55	ProteinTech	Cat: 24868-1-AP	AB_2879766	24868-1-AP is XBP1S-specific Fusion Protein expressed in *E. coli*
anti-Rbt HRP secondary	1:500–1:2500	goat	N/A	CST	Cat: 7074	AB_2099233	

**Table 2 ijms-24-09831-t002:** Test statistics.

Measure	Comparison	Test Statistic	*p* Value
Results Section 2.1 Weight, Morbidity, & Mortality following TBI in adolescent mice			
Weight	2-Way ANOVA Main effect of Injury	F_2,221_ = 5.95	0.009
	2-Way ANOVA Main Effect of Day	F_8,221_ = 62.38	<0.001
	2-Way ANOVA Interaction	F_16,221_ = 23.28	<0.001
Activity	3-Way ANOVA Main effect of Injury	F_2,65_ = 5.99	0.005
	3-Way ANOVA Main Effect of Day	F_10,65_ = 3.02	0.004
	3-Way ANOVA Main Effect of Light	F_1,65_ = 163.41	<0.001
Mortality x Time in Oxygen	Chi Square	X^2^ = 10.47	0.001
Agonal Breathing	Student’s *t*-test	t = 0.22	0.82
	Mean time TBI room air	1.3 min	
	Mean time TBI + O_2_	1.4 min	
Righting Time	Student’s *t*-test	t = 1.48	0.16
	Mean time TBI room air	11.2 min	
	Mean time TBI + O_2_	11.4 min	
Oxygen x Righting Time	Correlation	r = 0.15	0.53
Mortality x Seizures	Chi Square	X^2^ = 12.22	0.007
Results Section 2.3. Oxygen slows acute RGC loss but does not ultimately prevent loss of RGCs.			
Retinal Cell Loss Day 7	One-Way ANOVA Peripheral	F_2,42_ = 14.89	<0.001
	One-Way ANOVA Mid-Peripheral	F_2,42_ = 4.71	0.01
	One-Way ANOVA Central	F_2,42_ = 12.84	<0.001
Retinal Cell Loss Day 30	One-Way ANOVA Peripheral	F_2_,_42_ = 12.01	<0.001
	One-Way ANOVA Mid-Peripheral	F_2,42_ = 14.58	<0.001
	One-Way ANOVA Central	F_2,42_ = 0.58	0.5
Results from Section 2.4. Oxygen exposure does not prevent degeneration but may accelerate recovery.			
FJ-C Degeneration	2-Way ANOVA Main effect of Injury	F_2,51_ = 52.66	<0.001
	2-Way ANOVA Interaction	F_2,52_ = 11.97	<0.001
Results from Section 2.5. Astroglial reactivity is significantly reduced in the brain following O_2_ exposure			
GFAP Mean Fluorescence Intensity Day 7	One-Way ANOVA	F_2,33_ = 97.4	<0.001
GFAP Mean Fluorescence Intensity Day 30	One-Way ANOVA	F_2,22_ = 4.27	0.03
Results from Section 2.6. Oxygen exposure promotes IRE-1α pathway activation acutely			
IRE1α 7 Day	One-Way ANOVA	F_2,20_ = 3.47	0.04
XBP1-U 7 Day	One-Way ANOVA	F_2,20_ = 0.75	0.5
XBP1s 7 Day	One-Way ANOVA	F_2,20_ = 0.79	0.47
XBP1s:XBP1-U 7 Day	One-Way ANOVA	F_2,20_ = 0.5	0.6
IRE1α 30 Day	One-Way ANOVA	F_2,19_ = 7.3	0.004
XBP1-U 30 Day	One-Way ANOVA	F_2,28_ = 1.5	0.24
XBP1s 30 Day	One-Way ANOVA	F_2,28_ = 6.39	0.006
XBP1s:XBP1-U 30 Day	One-Way ANOVA	F_2,28_ = 7.68	0.002
Results from Section 2.7. The PERK pathway is sub-acutely elevated after TBI, but supplemental oxygen reduces long-term expression of pro-apoptotic markers.			
PERK 7 Day	One-Way ANOVA	F_2,21_ = 0.66	0.5
p-PERK 7 Day	One-Way ANOVA	F_2,21_ = 0.22	0.8
PERK:p-PERK 7 Day	One-Way ANOVA	F_2,21_ = 2.35	0.12
eIF2α 7 Day	One-Way ANOVA	F_2,21_ = 0.52	0.6
p-eIF2α 7 Day	One-Way ANOVA	F_2,21_ = 0.52	0.6
eIF2 α:p-eIF2α 7 Day	One-Way ANOVA	F_2,23_ = 2.8	0.08
ATF4 7 Day	One-Way ANOVA	F_2,21_ = 2.87	0.08
p-ATF4 7 Day	One-Way ANOVA	F_2,21_ = 6.24	0.007
ATF4:p-ATF4 7 Day	One-Way ANOVA	F_2,22_ = 6.37	0.007
CHOP 7 Day	One-Way ANOVA	F_2,15_ = 7.55	0.007
ERO1Lα 7 Day	One-Way ANOVA	F_2,23_ = 0.67	0.52
PERK 30 Day	One-Way ANOVA	F_2,33_ = 2.71	<0.001
p-PERK 30 Day	One-Way ANOVA	F_2,32_ = 0.14	0.87
PERK:p-PERK 30 Day	One-Way ANOVA	F_2,28_ = 1.56	0.23
eIF2α 30 Day	One-Way ANOVA	F_2,31_ = 7.9	0.002
p-eIF2α 30 Day	One-Way ANOVA	F_2,30_ = 11.21	<0.001
eIF2 α:p-eIF2α 30 Day	One-Way ANOVA	F_2,29_ = 2.36	0.11
ATF4 30 Day	One-Way ANOVA	F_2,17_ = 1.46	0.26
p-ATF4 30 Day	One-Way ANOVA	F_2,17_ = 3.2	0.06
ATF4:p-ATF4 30 Day	One-Way ANOVA	F_2,19_ = 1.12	0.35
CHOP 30 Day	One-Way ANOVA	F_2,8_ = 9.4	0.18
ERO1Lα 30 Day	One-Way ANOVA	F_2,18_ = 4.19	0.03

## Data Availability

All data are available upon request to the corresponding author.

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
