# Peer review of "Brief Oxygen Exposure after Traumatic Brain Injury Hastens Recovery and Promotes Adaptive Chronic Endoplasmic Reticulum Stress Responses"

_ijms, 2023, doi:10.3390/ijms24129831_

Round 1

Reviewer 1 Report

This is an amazing work with a huge selection of animals. The article uses modern methods. All results are presented in an excellent way, the voluminous supplement contributes to the understanding of the work done. 1) The association between brain oxygen deprivation (i.e., hypoxia) and traumatic brain injury (TBI) has been known for decades[1] with a reported co-incidence of up to 44%.[2] The link [XXX] should be immediately put, and then a dot and the end of the sentence. It needs to be corrected throughout the text.

2) I would recommend transferring Figure 1 to the material and methods. In the legend to Figure 1, it is necessary to give a decoding of the abbreviations. If the authors want to keep Figure 1 in the results, then text describing procedures and manipulations with animals is required.

3) Figure 7. The spelling of protein names should be the same

4) The discussion in the article is very well structured. It is a pleasure to read this section of the work. It may be worth discussing reactive astrogliosis, its two types, and its association with ER stress.

5) If the authors at the end of their article present a summarizing figure based on the materials received, then the work would become even better. Optional.

I am not a native English speaker. I didn't find any serious errors. The article is easy to read and understand.

Author Response

1) The association between brain oxygen deprivation (i.e., hypoxia) and traumatic brain injury (TBI) has been known for decades[1] with a reported co-incidence of up to 44%.[2] The link [XXX] should be immediately put, and then a dot and the end of the sentence. It needs to be corrected throughout the text.

Point 1: We have made the necessary adjustments throughout the manuscript (see tracked changes).

2) I would recommend transferring Figure 1 to the material and methods. In the legend to Figure 1, it is necessary to give a decoding of the abbreviations. If the authors want to keep Figure 1 in the results, then text describing procedures and manipulations with animals is required.

Point 2: We have moved Figure 1 to the materials and methods section. Figure legends have also been changed to detail abbreviations used and the order/naming of all other figures updated to reflect changes. 

3) Figure 7. The spelling of protein names should be the same

Point 3: This error has been fixed and the figure placed back into manuscript. As per point #2, this figure is now figure #6.

4) The discussion in the article is very well structured. It is a pleasure to read this section of the work. It may be worth discussing reactive astrogliosis, its two types, and its association with ER stress.

Point 4: We appreciate this compliment and suggestion and have added a few sentences on the dual nature of astrocytes (lines 403-414). However, we did not examine ER stress in astrocytes (or brain regions), nor did we report gliosis in the retina where we performed our measures of ER stress. Thus, because the discussion is already quite lengthy and this association between astrocytes and ER stress is beyond the scope of our analyses, we did not include this aspect in our discussion. 

5) If the authors at the end of their article present a summarizing figure based on the materials received, then the work would become even better. Optional.

Point 5: We created a new figure to summarize the results. It is now figure 11, next to the conclusion.

Reviewer 2 Report

The article “Brief oxygen exposure after traumatic brain injury speeds recovery and promotes adaptive chronic endoplasmic reticulum 3 stress responses “targets mainly the effects of oxygen exposure in recovering after TBI and its subsequent mortality. It’s a very thoroughly written paper, leaving the reader very few data to argue against, I will present some as it follows:

1.      Overall English grammar is fine but there are a few mistakes especially in the Results part (eg L113-115).

2.      In Results you are talking about seizures in mice and used graphic to show its results (Figure 2D). I believe there should be more information presented about the seizures, how were they evaluated etc.

3.      Lines 67-69 should be rephrased as they are hard to follow.

4.      Results - perhaps the amount of statistical data presented throughout the text should be organized in tables as it would make it easier to understand.

Author Response

1.Overall English grammar is fine but there are a few mistakes especially in the Results part (eg L113-115).

Point 1: We have fixed the grammar and run-on sentences in lines 113-115 as well as rewording other text throughout the results (see track changes).

2.In Results you are talking about seizures in mice and used graphic to show its results (Figure 2D). I believe there should be more information presented about the seizures, how were they evaluated etc.

Point 2: We have elaborated on how seizures were evaluated. See tracked changes in lines 89-91 for changes.

3.Lines 67-69 should be rephrased as they are hard to follow.

Point 3: This has been fixed with the addition of the results table 2 placed in the materials and methods section.

4. Results - perhaps the amount of statistical data presented throughout the text should be organized in tables as it would make it easier to understand.

Point 4: The test statistics have been removed from the results throughout the manuscript body. This comment has been addressed by creating an organized table of all statistical results/test statistics. This can now be easily found in Table 2 in materials and methods.